# GSDM-WBT: Global station-based daily maximum wet-bulb temperature data for 1981-2020

Jianquan Dong[1, 2], Stefan Brönnimann[2], Tao Hu[1], Yanxu Liu[3], Jian Peng[1, *]

[1]Laboratory for Earth Surface Processes, Ministry of Education, College of Urban and Environmental Sciences, Peking University, Beijing, 100871, China
[2]Institute of Geography and Oeschger Centre for Climate Change Research, University of Bern, Bern, 3012, Switzerland
[3]State Key Laboratory of Earth Surface Processes and Resource Ecology, Faculty of Geographical Science, Beijing Normal University, Beijing 100875, China

*Correspondence to*: Jian Peng (jianpeng@urban.pku.edu.cn)

**Abstract.** The wet-bulb temperature ($T_W$) comprehensively characterizes the temperature and humidity of the thermal environment and is a relevant variable to describe the energy regulation of the human body. The daily maximum $T_W$ can be effectively used in monitoring humid heatwaves and their effects on health. Because meteorological stations differ in temporal resolution and are susceptible to non-climatic influences, it is difficult to provide complete and homogeneous long-term series. In this study, based on the sub-daily station-based dataset of HadISD and integrating the NCEP-DOE reanalysis dataset, the daily maximum $T_W$ series of 1834 stations that have passed quality control were homogenized and reconstructed using the method of Climatol. These stations form a new data set of global station-based daily maximum $T_W$ (GSDM-WBT) from 1981 to 2020. Compared with other station-based and reanalysis-based datasets of $T_W$, the average bias was -0.48°C and 0.34°C respectively. GSDM-WBT handles stations with many missing values and possible inhomogeneities, and also avoid the underestimation of the $T_W$ calculated from reanalysis data. The GSDM-WBT dataset can effectively support the research on global or regional extreme heat events and humid heatwaves. The dataset is available at https://doi.org/10.5281/zenodo.7014332 (Dong et al. 2022).

## 1 Introduction

The trend of warming is threatening the climate system, terrestrial and marine ecosystems, and socio-economic development, resulting in the increase of the frequency and intensity of extreme climatic events, loss of biodiversity and protected areas, and human morbidity and mortality (Sun et al., 2014; Perkins-Kirkpatrick et al., 2020). Long-term temperature datasets have become the basis for accurate assessment of global or local warming and its impacts, especially heatwaves and their effects on health (Doutreloup et al., 2022; Fang et al., 2022). Previous studies on extreme heat mostly use near-surface air temperature directly based on observations from meteorological stations or numerical climate simulations (Mazdiyasni et al., 2017; Dong et al., 2021; Fischer et al., 2021), but the intensity of air temperature is usually not equivalent to the human body's response to the thermal environment. Human thermal comfort is related to many climatic and non-climatic conditions such as air

temperature, humidity, air pressure, skin albedo and heat insulation of clothing. For example, extreme humid heat combining with low air temperature but a high humidity might still cause lethal and even deadly events (Mora et al., 2017; Raymond et al., 2020). Indicators such as wet-bulb temperature ($T_W$) (Ahmadalipour and Moradkhani, 2018), apparent temperature (Hu and Li, 2020), humidex (Ho et al., 2017), and universal thermal climate index (UTCI) (Di Napoli et al., 2018) were proposed

to characterize thermal comfort of human bodies. Among them, $T_W$ has clear thermodynamic properties, and the higher $T_W$ could dampen the evaporative cooling of sweating (Kang and Eltahir, 2018). $T_W$ has been widely applied to multi-scale research on humid heat stress due to the mature methods (Pal and Eltahir, 2016; Raymond et al., 2020; Zhang et al., 2021). For example, Yu et al. (2021) found that in arid regions of Eurasia, changes of $T_W$ had stronger dependence on relative humidity than that in humid regions, and an increase of 1% in relative humidity would result in an increase of 0.2°C in $T_W$.

Near-surface air temperature and humidity are the key variables for calculating $T_W$ (Im et al., 2017). Although reanalysis and modelling datasets have the advantages of diverse parameters and complete series, studies have shown that changes in $T_W$ might be underestimated (Freychet et al., 2020). In comparison, station-based datasets are more difficult to provide continuous and homogeneous data, because meteorological observations can be directly or potentially affected by the damage of instruments, the relocation of stations, and also the surrounding environmental changes (Mamara et al., 2013; Li et al., 2020).

There is still a lack of public, downloadable global station-based datasets of $T_W$, especially for long-term series of daily maximum $T_W$ which can be used for research on extreme humid heat. In addition, another difficulty in generating station-based datasets of daily maximum $T_W$ is the impact of the temporal resolution of source data on the accuracy, because the daily maximum $T_W$ is not necessarily corresponding to the daily maximum temperature and daily maximum or minimum humidity. When only the daily-scale data are available, it often has to use daily average $T_W$ instead of calculating the real maximum

values (Yu et al., 2021; Guo et al., 2022). With the enhancement of continuity and resolution of data sources, hourly or sub-daily $T_W$ can be computed firstly, and then the daily maximum $T_W$ is obtained statistically (Im et al., 2017; Speizer et al., 2022).

HadISD, a sub-daily climatic dataset widely used in recent years, contains a set of basic meteorological variables, and it has also developed one humidity dataset and one heat stress dataset (Dunn et al., 2016). The humidity dataset of HadISD (HadISD-Humidity) includes $T_W$ data calculated from empirical formulas. Many studies used an algorithm proposed by

Davies-Jones to calculate $T_W$ (Davies-Jones, 2008), which allows to use such climatic variables as near-surface air temperature, humidity, and air pressure in HadISD. However, $T_W$ calculated in this way cannot deal with missing values and inhomogeneities. Although producers of HadISD provide a homogeneity assessment for temperature, dew point temperature, sea level pressure and wind speed (Dunn et al., 2014), the results are mostly used for quality control to assess their suitability for different research objectives. To our knowledge, there is no dataset that contains long-term complete series of daily

maximum $T_W$ based on global stations.

To this end, we used the HadISD sub-daily data and integrated reanalysis data to produce a global station-based daily maximum $T_W$ (GSDM-WBT) dataset, which spans 40 years (1981-2020) for 1834 stations (Dong et al. 2022). The GSDM-WBT solved the problems of many missing values and prominent inhomogeneity through data quality control and homogenization. We also evaluated the series of GSDM-WBT by comparing with the HadISD-Humidity dataset, as well as

another reanalysis-based dataset. The GSDM-WBT could provide data support for global or regional analysis (especially in the middle and high latitudes of the Northern Hemisphere) on long-term humid heat.

## 2 Methods

The production of GSDM-WBT includes four procedures: the calculation of $T_W$, data quality control, homogenization, and comparison and evaluation (Fig. 1). Specifically, based on the initial data of near-surface air temperature, specific humidity
and station level air pressure from HadISD, the algorithm proposed by Davies-Jones was used to calculate the sub-daily $T_W$. Further, by defining the valid days and valid months for the long-term series of $T_W$, the data quality was controlled and the daily maximum $T_W$ was obtained for valid stations. The homogenization was carried out in different station zones divided by the Köppen-Geiger climate classification, and reanalysis data were integrated to complement the series. In this part, the method of Climatol was used to correct inhomogeneous series and infill all missing values. Finally, we compared the differences
between the GSDM-WBT and other station-based and reanalysis-based datasets for better validating the accuracy.

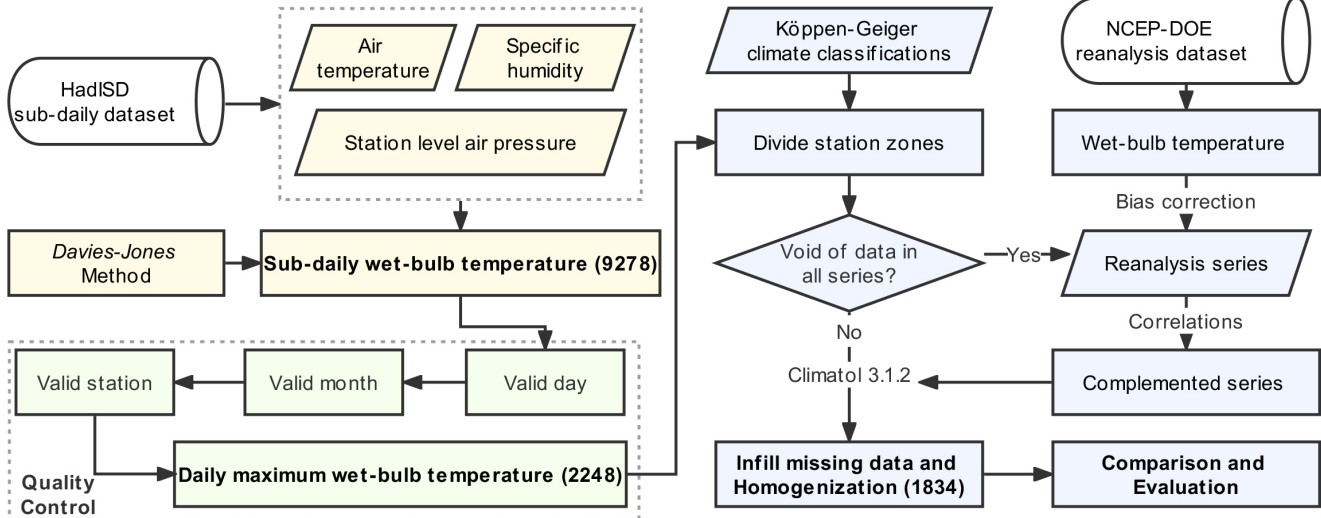

**Figure 1.** Procedures of producing global station-based daily maximum wet-bulb temperature (GSDM-WBT) dataset. The numbers in the parentheses indicate the counts of stations remained after each procedure.

## 2.1 Data sources

The HadISD was used to provide basic data of different climatic variables for GSDM-WBT. HadISD, launched by the Met Office Hadley Centre, uses station-based dataset from the Integrated Surface Database (ISD) (Smith et al., 2011) and is quality-controlled, with particular preservation of historical extreme values for meteorological variables. At present, the dataset has covered the observed data of more than 9,000 meteorological stations around the world. The time series can be traced back to 1931, and the temporal resolution is from one hour to daily scale (Dunn et al., 2016). Based on the algorithm of calculating

$T_W$, the near-surface (2m) air temperature (°C), specific humidity (g/kg), and station level air pressure (hPa) from 1981 to 2020 were imported. The used version of HadISD is v3.2.0.2021f. Considering the dependence of the occurrence of maximum $T_W$ at sub-daily scale on local climate, we converted Universal Time Coordinated (UTC) to the local time zone of each station.

Köppen-Geiger climate classification data were used for dividing station zones before homogenization. The "Present-day" climate classification was derived based on the monthly temperature and precipitation from 1980 to 2016, which included

three levels and was produced to three resolutions (Beck et al., 2018). Considering the density of stations in this study, the second-level with moderate resolution (0.083°) climate classification was selected, including 13 classes as Tropical-Rainforest, Tropical-Monsoon, Tropical-Savannah, Arid-Desert, Arid-Steppe, Temperate-Dry summer, Temperate-Dry winter, Temperate-Without dry season, Cold-Dry summer, Cold-Dry winter, Cold-Without dry season, Polar-Tundra, and Polar-Frost.

NCEP-DOE reanalysis dataset was used for complementing series in homogenization. NCEP-DOE is the second-

generation assimilated historical dataset produced by the National Oceanic and Atmospheric Administration of U.S. (Kanamitsu et al., 2002). The NCEP-DOE reanalysis reaches back to 1979 and provides 4 times daily values of various climate variables as well as daily and monthly means. The series of 2m air temperature (K), 2m specific humidity (kg/kg), and surface pressure (Pa) from 1981 to 2020 were used to calculate the sub-daily $T_W$ and daily maximum $T_W$, and linear scaling was used to correct the reanalysis series (Shrestha et al., 2017).

**2.2 Calculate the $T_W$**

The algorithm of calculating $T_W$ proposed by Davies-Jones has low error and is widely used (Raymond et al., 2020; Rogers et al., 2021). Based on the empirical formula for accurate calculation of equivalent potential temperature proposed by Bolton in 1980, Davies-Jones put forward the relationship among $T_W$, saturated mixing ratio, saturated vapor pressure and equivalent temperature. When an initial $T_W$ is given, the converged $T_W$ could be obtained by iterative calculation. The core formula is as

follows:

$$\left(\frac{C}{T_E}\right)^\lambda = f(T_W, \pi) \equiv \left(\frac{C}{T_W}\right)^\lambda \left[1 - \frac{e_s(T_W)}{p_0 \pi^\lambda}\right]^{\lambda v} \pi^{-\lambda k_3 r_s(T_W, \pi)} \exp\left[-\lambda G(T_W, \pi)\right] \qquad (1)$$

$$\tau_{n+1} = \tau_n - \frac{f(\tau_n; \pi) - \left(\frac{C}{T_E}\right)^\lambda}{f'(\tau_n; \pi)} \qquad (2)$$

Where $k_3$ and $v$ are the empirical parameters proposed by Bolton (Bolton, 1980), which are 0 and 0.2854, respectively. $T_E$ and $T_W$ are equivalent temperature and wet-bulb temperature. $e_s$, $r_s$ and $\pi$ are saturation vapor pressure, saturation mixing

ratio and nondimensional pressure. $C$, $\lambda$ and $p_0$ are constants, which are 273.15 K, 3.504 and 1000 mb respectively. $\tau_n$ and $\tau_{n+1}$ are the $T_W$ after the $n^{th}$ and $n+1^{th}$ iterations, and $\tau_n$ is set as the initial $T_W$ at the first iteration. Davies-Jones also showed the calculation of initial $T_W$ (Davies-Jones, 2008). When the equivalent temperature is in the ranges of high values or low values, the relationship between $T_W$ and $(\frac{C}{T_E})^\lambda$ is non-linear, and otherwise there is a linear relationship.

We referred to Buzan's implementation and Kopp's Matlab code to calculate $T_W$, and the threshold of convergence or the

maximum number of iterations were set to 0.001K and 100 respectively (Buzan et al., 2015; Kopp, 2020). Air temperature

(°C), specific humidity (kg/kg) or relative humidity (%), and air pressure (hPa) are input variables, and $T_W$ (°C) is the output variable. Specifically, long-term series of air temperature and humidity at sub-daily scale were directly imported, and the long-term average air pressure was used as a substitute because many observations of station level air pressure were missing. We performed the sensitivity analysis on comparing the differences in $T_W$ calculated using sub-daily air pressure and long-term average air pressure (Sect. 3.1 for details).

## 2.3 Data quality control

Due to the differences in temporal resolutions and the number of missing values among stations, it is necessary to conduct quality control of the original series in order to avoid extreme distribution of sub-daily $T_W$ and few valid data when calculating daily maximum $T_W$ (Zhang et al., 2021). Several criteria for data quality control were defined for better selecting valid stations:

I. Valid day: at least one $T_W$ every six hours (0-5 h, 6-11 h, 12-17 h, 18-23 h in local time) per day. Generally, the highest $T_W$ occurs in the daytime. However, because of the different temporal resolutions among stations or the inconsistent number of observations on different days at one station for HadISD, observations might only refer to extreme low values at night, thus resulting an underestimation of the daily maximum $T_W$.

II. Valid month: at least 21 valid days (three weeks) per month. Due to the high variability of daily data for long-term series, monthly series are often used as the basic data to correct daily series. For example, in the homogenization of daily temperature, it is first necessary to detect break points for the monthly series. If many valid days are missing in a month, it might cause a higher statistical deviation at the monthly scale.

III. Valid station: at least 400 valid months (of a total of 480 months during 1981-2020) per station. Considering the time span of 40 years, and hoping that the dataset could be useful for long-term research on extreme humid heat, we selected the stations which contain more valid months. It should be noted that here we do not require the selected stations to meet the definition of valid month in all 480 months, which is limited by the quality of data source. But further complementing series and infilling missing data could make up for this problem to a certain extent.

According to the above criteria, we screened out 2248 valid stations (Fig. S1), and computed the series of daily maximum $T_W$ for each station.

## 2.4 Homogenization

Homogenization is the key procedure which first detects the break points of long-term series caused by the influences of non-climatic factors (e.g., relocation of stations and surrounding environmental changes), and then corrects the data before and after the break points to improve the homogeneity of whole series (Brugnara et al., 2019; Fioravanti et al., 2019). The generally recognized process of correcting daily series was adopted, that is, firstly detecting break points at the monthly scale (480 time-steps in this study), and then correcting the daily series (14610 time-steps). Since it is difficult to obtain accurate historical information of stations, a relatively homogeneous reference series are often constructed from the data of stations surrounding

the candidate station. The break points could be identified through comparing whether there are significant differences between reference and candidate series.

### 2.4.1 Dividing station zones

The surrounding stations used to construct the reference series should have similar climatic backgrounds with the candidate station (Gubler et al., 2017), so as to ensure that the constructed reference series could be effectively used for detecting break points, especially in the context of large number of stations at the large scale. According to the second-level Köppen-Geiger climate classification at moderate resolution, there are 13 climate classifications in the world. As for 2248 valid stations selected after quality control, we divided them into several station zones based on climate classifications in ArcGIS 10.4, and
then the homogenization was performed in each station zone. In addition, in order to ensure that there were sufficient surrounding stations used to construct reference series, we required that there were at least 5 stations in each station zone, and finally got 41 station zones containing 1834 meteorological stations (Fig. S2).

### 2.4.2 Complementing series

Whether the reference value could be estimated for each time step of candidate station depends on how many missing data of
surrounding stations at this step. When all surrounding stations lack data, the estimation cannot be completed. Therefore, when the above situation arose, we introduced the reanalysis series as the complementary series to achieve homogenization for the candidate station. The NCEP-DOE reanalysis dataset also includes air temperature, specific humidity, and surface pressure every 6 hours from 1980 to 2020, but it might be affected by systematic and random errors, leading to the deviations from actual observations (Yan et al., 2020). A total of 36 station zones (except for Z13, Z19, Z25, Z26 and Z29) needed to be
supplemented by reanalysis series in this study. Firstly, the air temperature, specific humidity and surface pressure of the grid point nearest to each station were extracted, and the sub-daily (six-hour interval) $T_W$ was calculated (see Sect. 2.2). Then the initial series of daily maximum $T_W$ and monthly mean were computed before bias correction. Furthermore, the linear scaling (Shrestha et al., 2017) was used to calculate the bias of the average monthly mean series between each station and the nearest grid point from January to December. Finally, the bias was used to correct the daily maximum $T_W$ of the nearest grid point for
each month. Equations are as follows:

$$TW_{max}(r)^* = TW_{max}(r) + [Mon_{mean}(s) - Mon_{mean}(r)] \qquad (3)$$

Where, $TW_{max}(r)$ and $TW_{max}(r)^*$ are the original and corrected series of daily maximum $T_W$ based on reanalysis data, respectively. $Mon_{mean}(s)$ and $Mon_{mean}(r)$ are the long-term average monthly mean series from station-based data and reanalysis-based data, respectively.

Due to the relatively coarse resolution of reanalysis dataset, one grid might involve two or more stations spatially. We deleted the duplicate series and paired it with the station-based series with the highest correlation coefficients for further bias correction. Besides, the number of complemented series is equal to the number of stations in such zones that should be supplemented theoretically, but too many complementary reanalysis data would reduce the reliability of constructing reference

series. After removing the duplicating series, reanalysis series which had the top 10% correlation coefficients ($p<0.05$) with station-based series were selected as complementary series for the corresponding station zone.

### 2.4.3 Infilling missing data and homogenization

Many algorithms of identifying inhomogeneity and homogenization have been proposed, such as MASH (Mamara et al., 2013), RHtests (Brugnara et al., 2020), HOMER (Coll et al., 2020), and Climatol (Dumitrescu et al., 2020). These algorithms differ in methods of detecting break points, applicable variables and their resolutions, the number of series to be processed, and the ability of automation. Climatol has the advantages of high tolerance for missing data, unlimited variables, and unlimited sample size. Climatol selects the reference stations according to the distance to candidate stations, estimates the reference series based on the reduced major axis regression, and then applies the Standard Normal Homogeneity Test (SNHT) to the series of anomalies between the actual values and the reference values to identify the break points (Alexandersson, 1986). Since SNHT is a method of detecting single break-point, Climatol conducts the detections on the stepped overlapping temporal windows and on the complete series respectively in order to avoid ignoring the multiple break points in the series. One inhomogeneous series can be divided into several homogenous sub-series. Finally, all missing data were infilled by averaging neighbouring values. Both infilling missing data and constructing reference series rely on data normalization, which might have high uncertainty when the series is incomplete. Climatol iteratively infills missing data multiple times until the mean of series becomes stable (Paulhus and Kohler, 1952). The procedures of Climatol are shown in Fig. S3.

In this study, Climatol (version 3.1.2) with an R script was used to perform homogenization in each station zone. Since Climatol selects the reference station based on the distances between stations and ignores the correlations of series, we calculated the average correlation coefficients of the candidate and the surrounding series with the increase of the number of reference stations in each station zone, and then selected the maximum number of reference stations as the imported parameter in Climatol (Sect. 3.2 for details). In addition, in the stage of infilling missing values, Climatol allows setting weights to surrounding stations, that is, the weights decay as the distances to the candidate station increase. In each station zone, the average distance between the candidate stations and the nearest stations was set as the distance parameter for half weight. In the stage of detecting break points, we firstly conducted exploratory experiments to obtain the standard deviation of the series and the frequency distribution of SNHT values, and then determined the thresholds for deleted outliers and break points (Table S1 for details on parameters). Higher standard deviations and SNHT values mean higher probability of such stations to be detected as the outliers and break points. Through setting the above parameters, we detected the break points for the monthly series of average daily maximum $T_W$, that is, set it as the known meta-data information, and then split the daily series and reconstructed series.

### 2.5 Sensitivity analysis

There are two potential uncertainties in the procedures of calculating $T_W$ and homogenization when producing GSDM-WBT. Firstly, due to the missing observations of station level air pressure, we assumed that the influence of air pressure on $T_W$ was

much lower than that of air temperature and humidity in the long-term state, and thus the long-term average air pressure was used instead of the sub-daily air pressure. We assessed the average bias of the daily maximum $T_W$ to check the effect of long-term average air pressure. Secondly, the important difference between the Climatol and other algorithms of homogenization is that the reference stations are selected based on their distances from the candidate stations rather than the correlation of series. Therefore, when setting the maximum number of reference stations, we also considered the changes of correlation between different numbers of reference stations and candidate stations.

## 3 Results

### 3.1 Effect of long-term average air pressure

To evaluate the effect of long-term average air pressure on the daily maximum $T_W$, we applied the same algorithm to calculate $T_W$ based on sub-daily air pressure, and also used the same criteria of data quality control to select 398 valid stations. The average bias of the daily maximum $T_W$ based on the long-term average and sub-daily air pressure for such 398 stations was 0.12°C. In view of spatial patterns (Fig. 2), arid and semi-arid regions had the clustering of high bias, and other mid-latitude regions had lower bias which was mostly concentrated at 0-0.15°C, whereas the bias increased in high-latitude regions. Sensitivity analysis of previous studies also showed that the effect of surface pressure on $T_W$ was about 0.1°C (Raymond et al., 2020). When targeting on the stations with average daily maximum $T_W$ more than 20°C, where humid heat conditions were highly relevant to human health, the average bias was also maintained at 0-0.11°C.

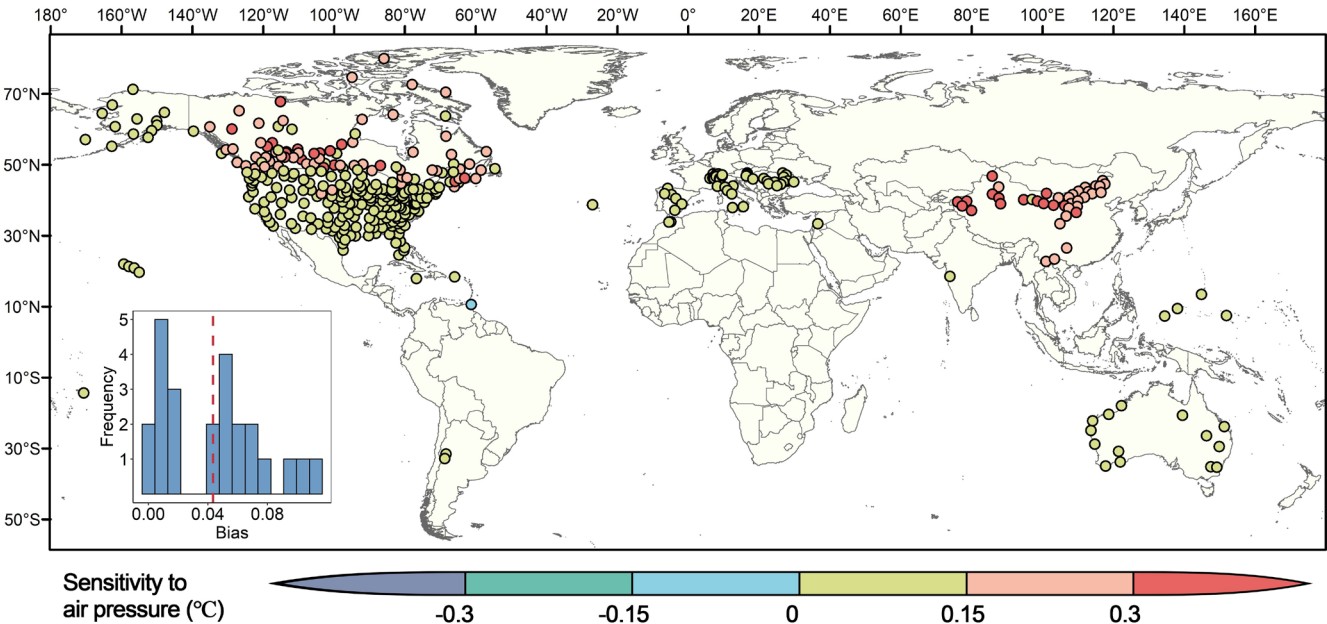

**Figure 2.** Sensitivity of $T_W$ to air pressure. Sensitivity, i.e., average bias, was calculated by subtracting the daily maximum $T_W$ calculated by sub-daily pressure from the daily maximum $T_W$ based on long-term average pressure. Sub-plot showed the
histogram of number of stations with corresponding average bias when average daily maximum $T_W$ was more than 20°C, where the red dashed line indicated the mean (0.04°C).

### 3.2 Correlation between candidate and reference stations

Before the homogenization, we calculated the changes of average correlation coefficients between the candidate series and surrounding series with the increase of the number of reference stations (Fig. 3). Stations that were closer to the candidate
stations were preferentially selected. Except for the Z32, Z33, Z35, Z36 and Z41 station zones, no matter how many reference stations are selected, the average correlation coefficients always remained above 0.9 (1789 stations in total). Ensuring a certain number of reference stations, the average correlation coefficients of Z32, Z33 and Z41 could be stable above 0.8, while Z35 and Z36 located near the equator have lower regional average coefficients. Therefore, it is emphasized that the GSDM-WBT might have higher reliability in mid-to-high latitudes.

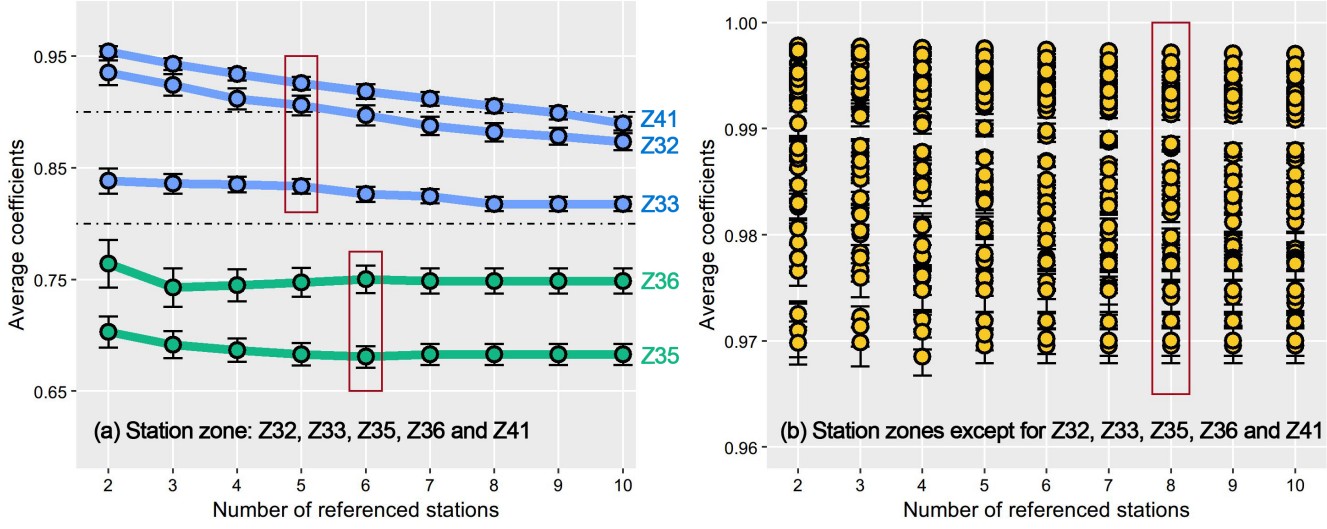


**Figure 3.** Average correlation coefficients between series of candidate and reference stations in different station zones. The red box highlights the number of maximum reference stations which was used for homogenization.

### 3.3 Effect of homogenization

Detection of inhomogeneity could identify the break points caused by non-climatic factors for long-term series. After
homogenization, in theory the corrected series of candidate stations should have a better correlation with the surrounding series. We paired 1834 stations and calculated the mutual correlation coefficients before and after homogenization (Fig. 4(a)). Overall, the correlation coefficients after correction were higher and the maximum increment of coefficients was 0.28. It was also notable that there was a significant increase in correlation between stations that were closer together as shown in the blue dots.

In the sub-plot of Figure 4 (a), about 80.23% of paired stations had larger coefficients after homogenizations. To further demonstrate the effect of homogenization, we selected one typical station from each station zone that either had the most break points, had higher SNHT values, or had more missing data (Table S2 for details). The changes of annual average daily maximum $T_W$ before and after the homogenization and the number of infilled and corrected data were shown in Z1-Z41 of Fig. 4. On the one hand, before the break points, some stations showed a significant increase or decrease in the average daily maximum $T_W$ before and after homogenization (e.g., Z2, Z8, Z18 and Z41). The overestimation or underestimation of the original series is mainly related to the equipment, environment and statistical methods of monitoring stations in different countries. On the other hand, many missing data directly lead to discontinuous series and abnormal statistical values. For example, a large number of missing values in the Z25 and Z29 station zones around 1995 caused abnormal fluctuations.

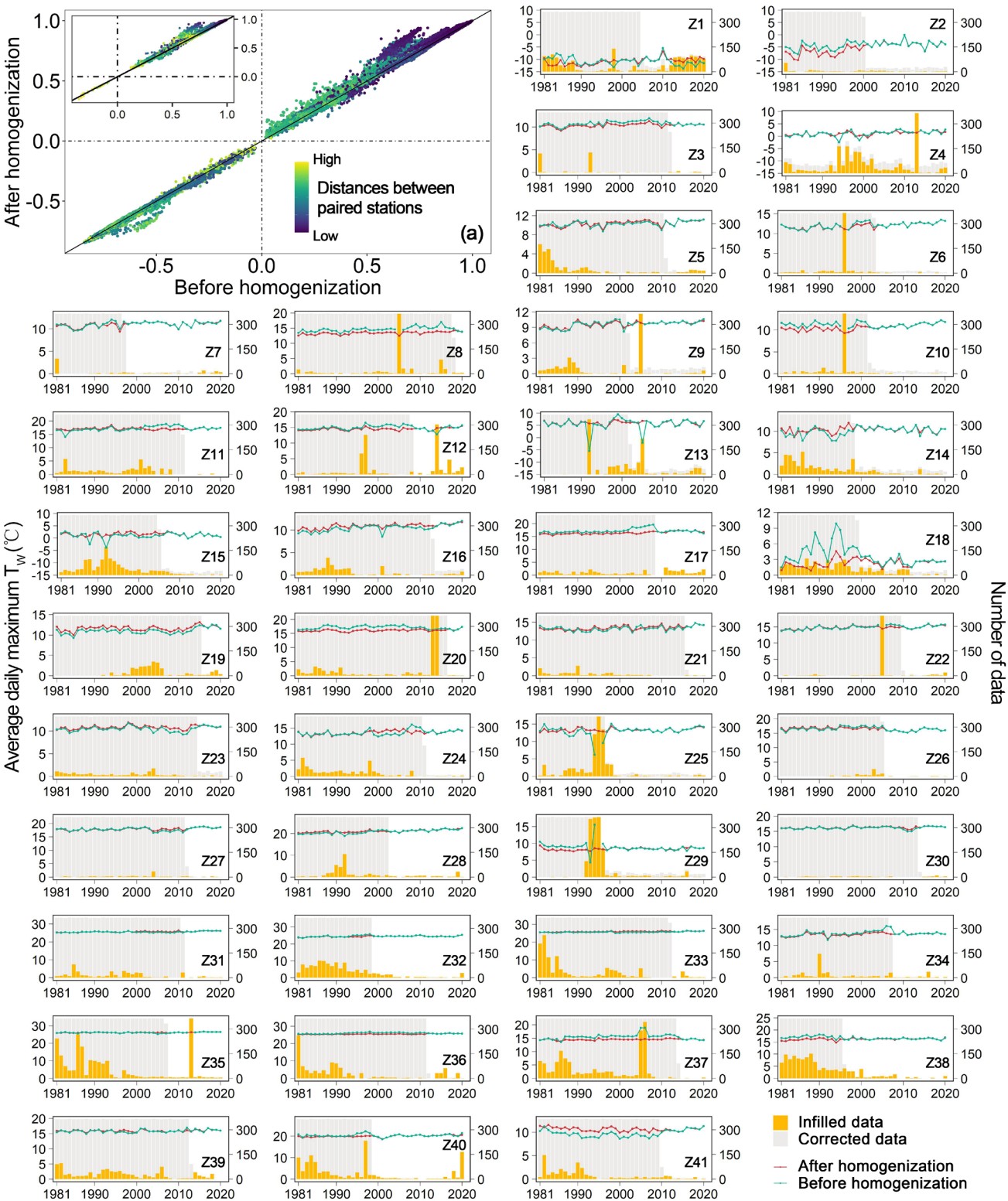

**Figure 4.** Correlation coefficients (p<0.05) between paired series before and after homogenization (a), and annual average daily maximum $T_W$ (°C) and the number of infilled or corrected data for one typical station in each station zone (Z1-Z41). Sub-plot of the figure (a) showed the correlation coefficients between paired stations of which distances lower than the first quarter. When the coefficients were more than 0, the dots in the upper areas of black diagonal indicated the higher coefficients after homogenization. Detailed information of all typical stations was shown in Table S2.

In addition, complementing series was an essential process to achieve all homogenizations, and reanalysis dataset was introduced in this study. To reduce the impact of uncertainty in the reanalysis data, we selected complementary series based on the correlation coefficients (Sect. 2.4) and also demonstrated the effect in different station zones as shown in Table S3. The number of complementary series was limited to no more than 10% of the number of all stations (at least one complementary series). The reanalysis-based dataset was mainly used to provide reference daily maximum $T_W$ when the values in each time step of all candidate stations were missing. However, such situation was not universal since the percentages of void time steps in series (0.03%-2.59%) relative to 14610 total time steps were quite low.

### 3.4 Evaluations

### 3.4.1 Comparison with station-based data

In addition to the basic meteorological variables, HadISD-Humidity also includes $T_W$ calculated by the simple empirical formulas. Since HadISD-Humidity directly uses the original dataset to calculate $T_W$ without further post-processing, it still has the shortcomings of many missing values and possible heterogeneity. We used the same definition to calculate the valid days for HadISD-Humidity, and counted the number of missing days in January-December during 1981-2020 for all 1834 stations. The median number of missing days in each month over past forty years in the Northern Hemisphere was less than 100 days, much lower than the corresponding months in the Southern Hemisphere (Fig. 5). In terms of seasonality, there were evidently more missing days in the warm season (May-September) in the Northern Hemisphere, especially in summer (June-August). Because the extremely humid heat events are generally identified based on daily $T_W$ and the daily thresholds in the historical baselines, more missing values could cause inaccurate thresholds or insufficient events to be detected. Therefore, the potential uncertainties should be noticed when directly using HadISD-Humidity to characterize humid heat.

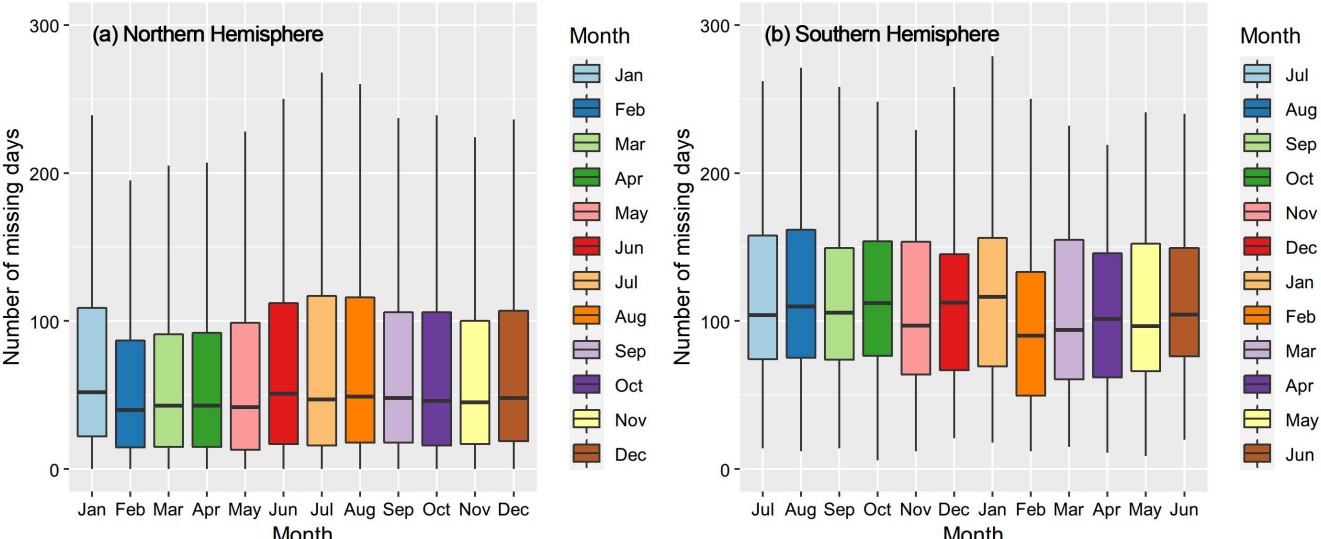

**Figure 5.** Number of missing days in different months during 1981-2020 for HadISD-Humidity dataset. The lower and upper hinges correspond to the 25th and 75th percentiles, and the horizontal lines in the boxes show the medians. The lower and upper whiskers are the minimum and maximum values.

The bias of daily maximum $T_W$ between GSDM-WBT and HadISD-Humidity was further calculated. Because the series
of $T_W$ from HadISD-Humidity were not corrected for homogeneity, the 1834 stations could not be fully matched. However, HadISD provides the test values of detecting inhomogeneity based on the pairwise homogenization algorithm (Menne and Williams, 2009), for the monthly mean diurnal range of air temperature and dew point temperature. Based on the detected results, 245 completely homogenous stations were screened out in this study from 1981 to 2020, which were concentrated in the middle latitudes (Fig. 6), although it is notable that the existing missing values might increase the potential inhomogeneity
of daily maximum TW series in HadISD-Humidity. Overall, the daily maximum $T_W$ of GSDM-WBT is lower than that of HadISD-Humidity. The mean of average bias for all stations was -0.48°C, and the average root mean square error (RMSE) was 0.72°C. In view of spatial patterns, western Europe had high consistency for these two datasets, and part stations in arid and semi-arid regions of central Asia and western North America had poor consistency.

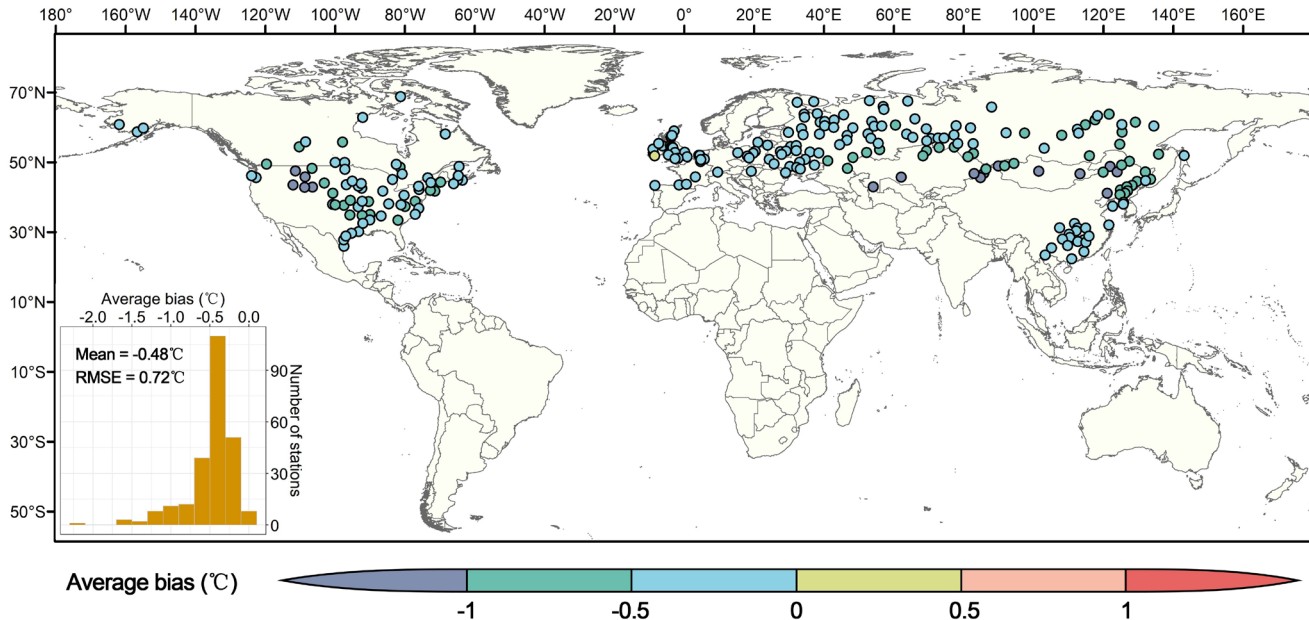

**Figure 6.** Average bias between daily maximum $T_W$ of GSDM-WBT and HadISD-Humidity.

### 3.4.2 Comparison with reanalysis-based data

ERA5 (Hersbach et al., 2020) has also been widely used in calculating various heat stress index and producing the corresponding dataset in recent years. Yan et al. (2021) launched a high-resolution thermal stress dataset (HiTiSEA) covering South and East Asia. The dataset with a spatial resolution of $0.1° × 0.1°$ and a time span of 1981-2019, includes daily maximum $T_W$. There are 587 stations of GSDM-WBT located in the spatial range of HiTiSEA. We extracted the HiTiSEA series of daily maximum $T_W$ in the nearest grid points to all 587 stations, and compared the average bias with GSDM-WBT (Fig. 7). Overall, compared with HiTiSEA, the means of average bias and RMSE for all stations were 0.34°C and 1.61°C respectively. High inconsistency between the two datasets existed in the north eastern and southern regions.

The verification of HiTiSEA showed that its average bias of the daily maximum $T_W$ from the meteorological stations was -0.4°C (Yan et al., 2021), which was consistent with our study. It should also be noted that HiTiSEA was produced from the sub-daily data of UTC, and thus we checked the correlation between the longitudes of stations and the average bias. The extremely low correlation coefficients indicated that the average bias was not dependent on longitude (local time zone) (Fig. S4).

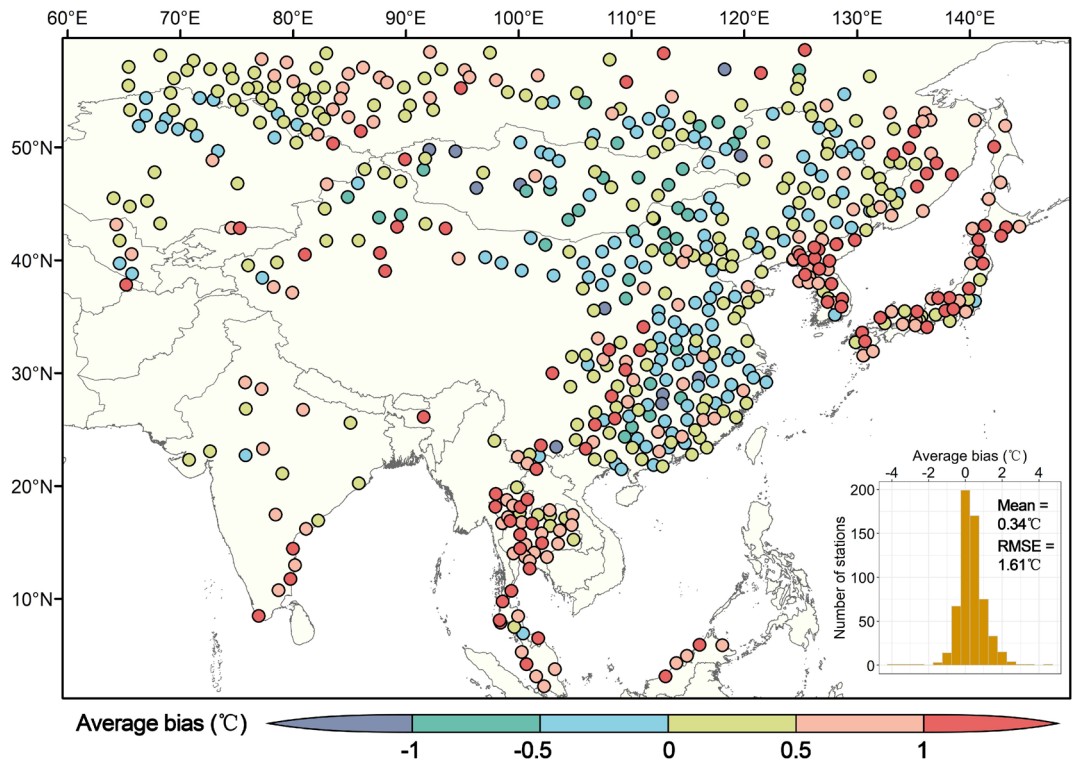

**Figure 7.** Average bias between station-based daily maximum $T_W$ of GSDM-WBT and that of the nearest grid points in HiTiSEA.

### 3.4.3 Year-to-year comparison

The annual average daily maximum $T_W$ was further calculated in 245 stations for the comparative analysis of GSDM-WBT and HadISD-Humidity, and in 587 stations for the comparative analysis of GSDM-WBT and HiTiSEA (Fig. 8). Overall, whether focusing on all months or only the warm season, the annual average daily maximum $T_W$ of GSDM-WBT was lower than that of station-based HadISD-Humidity, but higher than that of reanalysis-based HiTiSEA. In view of the relative accuracy, the former inconsistency may be caused by the existing missing values of HadISD-Humidity and the homogenization of GSDM-WBT. The latter differences have reached a similar conclusion in previous studies, that is, the $T_W$ and other heat stress indices calculated from reanalysis-based data are underestimated.

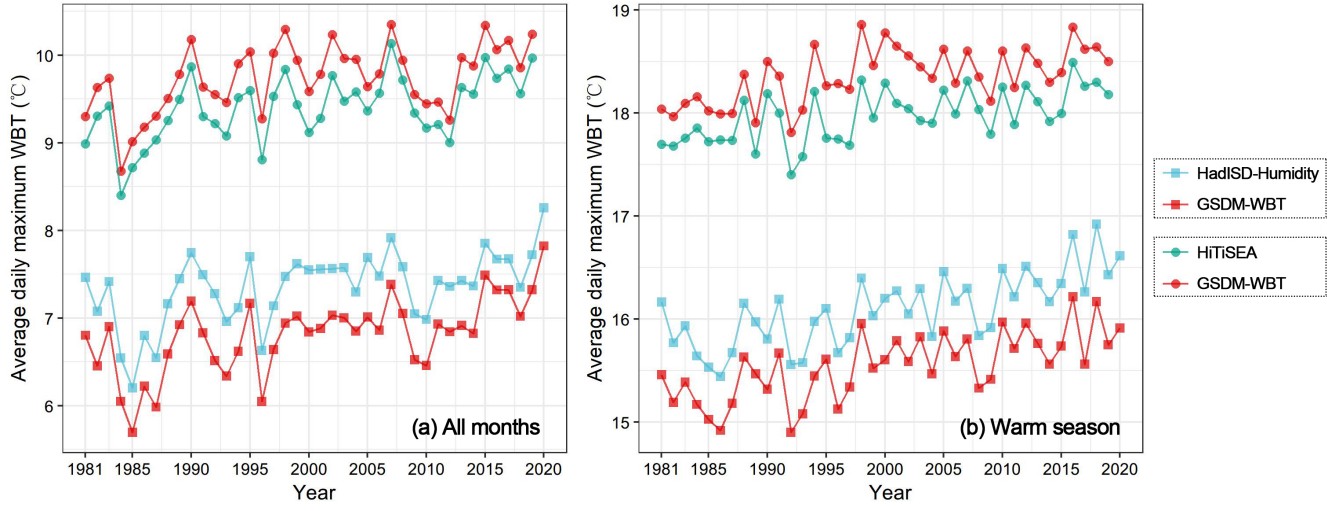

**Figure 8.** Annual average daily maximum $T_W$ between HadISD-Humidity, HiTiSEA and GSDM-WBT in all months and warm season (May, June, July, August and September in the Northern Hemisphere).

## 4 Discussion

### 4.1 Advantages of GSDM-WBT in climate change research

$T_W$, a characteristic temperature that integrates temperature and humidity, reflects the response of human bodies to the thermal environment and has been widely used in the fields of heat waves, climate and health, and social vulnerability (Coffel et al., 2018; Kang and Eltahir, 2018). Although $T_W$ is suitable for large-scale applications, there is still a lack of long-term datasets based on meteorological stations. Based on the observed data of HadISD and integrating reanalysis data, we have produced a dataset of daily maximum $T_W$ from 1981 to 2020 for 1834 stations around the world, which can effectively support global or

regional research on climate change and its impact. Two main advantages of GSDM-WBT should be emphasized. Firstly, compared with other thermal comfort indices, the algorithm of computing $T_W$ is relatively mature, and the required data sources are not complicated. The UTCI is also one typical thermal comfort indicator that has been gradually recognized in recent years, because it not only considers more climatic variables such as temperature, wind speed and humidity, but also considers the parameters of skin albedo and clothing conditions (Wang and Yi, 2021). However, the complete model of UTCI has high

complexity, and the existing research mainly uses the approximate polynomial fitting method. In addition, UTCI is mostly performed at small scales (Dong et al., 2020), while the localized parameters of UTCI are still difficult to obtain.

    Another advantage of GSDM-WBT is that Climatol was applied to achieve homogenization for daily maximum $T_W$, thereby eliminating the possible break points affected by non-climatic factors, and reconstructing the series without missing values. Although HadISD dataset has been used to compute $T_W$ in previous analysis of humid heat, such research either usually

ignored the inhomogeneity and missing values, or selected fewer stations by improving quality control (Zhang et al., 2021).

Therefore, the complete series reconstructed by GSDM-WBT can better serve the daily-scale research on thermal environment. For example, if there are many missing days, a continuous heatwave event would be divided into multiple independent events, and the cumulative intensity and duration of the heatwave might be underestimated. In addition, more accurate extreme values at the daily scale can be obtained based on sub-daily data sources. Previous research showed the differences of extreme humid heat between using monthly and sub-daily temperature and humidity could be up to more than 4°C at regional scale, and lead to substantial uncertainty of future predictions (Buzan and Huber, 2020). Different from the evaluations of extreme heat events in view of the average temperature, the daily maximum $T_W$ of GSDM-WBT better shows the real extreme thermal situation for one day.

## 4.2 Limitations and future improvements of dataset

Homogenization is an important procedure in the production of GSDM-WBT. Generally, detection of inhomogeneity is often applied to observed climate variables such as temperature, humidity and wind speed (Azorin-Molina et al., 2016; Li et al., 2020). Furthermore, it has also been applied in recent years for non-traditional meteorological variables such as plant phenology (Brugnara et al., 2020). We adopted the idea of calculating the $T_W$ first and then performing homogenization, but inevitably, the calculation of $T_W$ might smooth the break points of original series. The ideal process is to first perform homogenizations on several single variables (i.e., air temperature, humidity, and air pressure) for $T_W$, and then to combine all homogeneous series to calculate the $T_W$. However, the complexity and uncertainty of such ideal process are difficult to estimate. On the one hand, the temporal resolution of univariates is at hourly or sub-daily scale. The resolution is higher, the operation time increases, and more missing values may lead to lower accuracy of interpolation. Besides, the detected break points of different univariates do not correspond completely. When the historical meta-data is lacking, it is difficult to judge whether there is a conflict in break points between all variables, and to make sure how to determine the thresholds used for homogenization. Therefore, we conducted the procedures of calculating the $T_W$ firstly and then completing the homogenization. In the future, with the improvement of data availability, mature algorithm and complete records, homogenous series of univariates could be obtained firstly, followed by the calculation of daily maximum $T_W$.

Recent studies have also attempted to use existing algorithms to perform homogenization on sub-daily or hourly series, although they are still carried out at small scale (Dumitrescu et al., 2020). This is mainly because high-resolution meteorological datasets with good quality always need multi-sectoral cooperation within countries or cities. In the future, with the enhancement of the global meteorological station networks and data records, the $T_W$ dataset with higher temporal resolution could be constructed, which could not only improve the accuracy of daily statistics, but also promote the research on the differences between daytime and nighttime for better characterizing humid heat and exploring potential mitigations. Meanwhile, the complex changes in the relationship, but not the simply fixed joint, between temperature and humidity, was investigated around different regions based on the multivariate analysis (Mckinnon and Poppick, 2020). Then the historical dataset of $T_W$ could be expanded to future longer periods based on observation-based relationship between temperature and humidity (Poppick and Mckinnon, 2020).

## 5 Data availability

The GSDM-WBT dataset was freely available at https://doi.org/10.5281/zenodo.7014332 (Dong et al. 2022). We provide the NetCDF files of GSDM-WBT for each station and one compressed file containing all data.

## 6 Conclusions

Based on HadISD station-based observations and integrating with the NCEP-DOE reanalysis data, the daily maximum $T_W$ of 1834 stations around the world was produced through the calculation of $T_W$, data quality control, infilling missing values and
385 homogenization. The GSDM-WBT covers the complete daily series of forty years from 1981 to 2020. The production with the application of Climatol successfully corrected the inhomogeneities of series caused by non-climatic factors, and also infilled all missing data to reconstruct complete series for each station. Compared with the existing public-downloaded station-based and reanalysis-based $T_W$ datasets, the overall average bias of GSDM-WBT was -0.48°C and 0.34°C, with the average RMSE of 0.72°C and 1.61°C, respectively. This new dataset can better support the studies on global or regional humid heat
events. We also hope that with the improvement of observations and reconstructed algorithms, the uncertainty of producing the dataset can be further reduced and a global station-based $T_W$ dataset with hourly resolution can be produced in the future.

## Author contribution

Jianquan Dong proposed the ideas, produced the datasets and performed the data analysis and visualization. Jianquan Dong prepared the manuscript with contributions from all co-authors. Stefan Brönnimann and Jian Peng supervised for the
395 production and revised the manuscript.

## Competing interests

The authors declare that they have no conflict of interest.

## Acknowledgements

We kindly thank Dr. Jose A. Guijarro for his helps and suggestions on using the Climatol, and thank Dr. Yuri Brugnara for the
400 constructive comments of homogenization. We are grateful for the support from the China Scholarship Council.

## Financial support

This research was financially supported by the National Natural Science Foundation of China (42130505).

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
