# Peer review of "GSDM-WBT: Global station-based daily maximum wet-bulb temperature data for 1981-2020"

_Earth System Science Data, 2022_

## Author Comment (AC2)

Details of the revisions and responses to **Reviewer 1** comments on the manuscript entitled '*GSDM-WBT: Global station-based daily maximum wet-bulb temperature data for 1981–2020*' (essd-2022-309)

We would like to thank the reviewer for your insightful and constructive comments that help to enhance the overall quality of our manuscript. Our responses (on a comment-by-comment basis) are indicated in blue text, and all updates to the original submission will be highlighted in the revised manuscript.

**Comments:** The wet-bulb temperature (WBT) comprehensively characterizes the temperature and humidity of the thermal environment and is a relevant variable to describe the energy regulation of the human body. The manuscript proposed a global station-based daily maximum wet-bulb temperature data for 1981-2020, which is an important and exciting study for such a field. The GSDM-WBT dataset can effectively support the research on global or regional extreme heat events and humid heat waves. I think the study is well analyzed and did not have more comments. although I think the author should polish the expression of the manuscript.

Response: Thank you for recognizing our new dataset and providing the comments. We have further polished our manuscript for more professional expression and better understanding.

---

## Author Response (AR1)

Details of the revisions and responses to all comments on the manuscript entitled '*GSDM-WBT: Global station-based daily maximum wet-bulb temperature data for 1981–2020*' (essd-2022-309)

We would like to thank all reviewers and the community member for their insightful and constructive comments that help to enhance the overall quality of our manuscript. Our responses (on a comment-by-comment basis) are indicated in blue text, and all updates to the original submission are shown in the track-changes manuscript. We welcome any further comments on the revised manuscript and are willing to make additional revisions if required.

Jian Peng, Dr.
On behalf of all the authors

**Reviewer 1**

**Comments:** The wet-bulb temperature (WBT) comprehensively characterizes the temperature and humidity of the thermal environment and is a relevant variable to describe the energy regulation of the human body. The manuscript proposed a global station-based daily maximum wet-bulb temperature data for 1981-2020, which is an important and exciting study for such a field. The GSDM-WBT dataset can effectively support the research on global or regional extreme heat events and humid heat waves. I think the study is well analyzed and did not have more comments. although I think the author should polish the expression of the manuscript.

Response: Thank you for recognizing our new dataset and providing the comments. We have further polished our manuscript for more professional expression and better understanding.

**Comments:** I really like what the authors have set out to do, to make a homogenised station-based dataset of T_w with complete temporal coverage. I know that this is a complex task, and they should be commended for doing so. They have validated their product against a reanalysis (which is as independent as one can get for these kinds of observations) and a similar station-based product which uses alternative methods.

My major comment relates to the use of the NCEP DoE reanalysis, and how sensitive this data product is to the choice of this (older) reanalysis. Most other comments suggest improvements for the figures, the readability or other minor clarifications.

Response: Thank you for recognizing our new dataset and providing the comments. All responses to your major and minor comments are as follows.

**Major Comment:**

Line 95 - The NCEP DOE reanalysis is a relatively old product, and a number of studies have found issues with it. There are more recent reanalyses available (albeit being usually larger, these are more complex to use). How do you think your results depend on the choice of reanalysis?

Response: In our study, the NCEP-DOE reanalysis dataset was introduced to complement series when the values in each time step of all candidate stations were missing. This process was essential but with relative low effect of reanalysis series on the whole results. As shown in the Methods of manuscript, we selected the reanalysis series which had the top 10% correlation coefficients ($p<0.05$) with station-based series to improve the reliability of reference series. We could also investigate the number/percentage of void time steps in all series in different station zones (Table R1), and it was found that the percentages (0.04%-2.59%) relative to 14610 total time steps were low. Therefore, the selection of NCEP-DOE and its complementary series affects the eventual results slightly. We have demonstrated the effect of reanalysis series in the Results in the revised manuscript.

Table R1. The effect of complementary series in different station zones.

| Station zone | Number of complementary series | Number of all stations | Number of void time steps in all stations | Percentage of void time steps (%) in all stations |
|---|---|---|---|---|
| Z1 | 1 | 5 | 27 | 0.18 |
| Z2 | 1 | 8 | 8 | 0.05 |
| Z3 | 1 | 9 | 14 | 0.10 |
| Z4 | 1 | 5 | 36 | 0.25 |
| Z5 | 5 | 54 | 12 | 0.08 |
| Z6 | 1 | 9 | 10 | 0.07 |
| Z7 | 9 | 87 | 5 | 0.03 |
| Z8 | 1 | 11 | 8 | 0.05 |
| Z9 | 1 | 12 | 378 | 2.59 |
| Z10 | 45 | 451 | 6 | 0.04 |
| Z11 | 2 | 20 | 8 | 0.05 |
| Z12 | 1 | 12 | 6 | 0.04 |
| Z14 | 3 | 31 | 15 | 0.10 |
| Z15 | 4 | 35 | 12 | 0.08 |
| Z16 | 1 | 8 | 24 | 0.16 |
| Z17 | 1 | 9 | 27 | 0.18 |
| Z18 | 4 | 41 | 7 | 0.05 |
| Z20 | 1 | 5 | 169 | 1.16 |
| Z21 | 2 | 18 | 9 | 0.06 |
| Z22 | 3 | 34 | 13 | 0.09 |
| Z23 | 19 | 187 | 6 | 0.04 |
| Z24 | 4 | 41 | 7 | 0.05 |
| Z27 | 6 | 64 | 12 | 0.08 |
| Z28 | 1 | 5 | 67 | 0.46 |
| Z30 | 6 | 56 | 7 | 0.05 |
| Z31 | 4 | 38 | 35 | 0.24 |
| Z32 | 1 | 12 | 46 | 0.31 |
| Z33 | 1 | 8 | 41 | 0.28 |
| Z34 | 4 | 36 | 5 | 0.03 |
| Z35 | 1 | 7 | 145 | 0.99 |
| Z36 | 1 | 7 | 123 | 0.84 |
| Z37 | 1 | 9 | 21 | 0.14 |
| Z38 | 1 | 8 | 21 | 0.14 |
| Z39 | 2 | 24 | 41 | 0.28 |
| Z40 | 1 | 9 | 32 | 0.22 |
| Z41 | 1 | 11 | 17 | 0.12 |

Besides, there are several reasons why the NCEP-DOE could be as a selection in this study, especially after considering the availability and data volume. Firstly, its temporal resolution is consistent with our criteria of data quality control (i.e., at least one $T_W$ every six hours per day).

In addition, although other reanalysis (e.g., ERA5-land hourly data) have higher temporal resolutions, users might be required to compute the input variables of calculating $T_W$ by themselves (e.g., calculate the humidity based on the dewpoint temperature). The NCEP-DOE directly provides the 2m air temperature (K), 2m specific humidity (kg/kg) and surface pressure (Pa) to calculate $T_W$, which is in concert with the use of HadISD. In previous research on heat-related events (Mora et al., 2017; Wang et al., 2021), NCEP-DOE was also widely applied.

**Minor Comments:**

Line 12 - the phrase "and the response on human health" doesnt quite read correctly for me. I see what youre trying to say, but that sentence doesnt flow well.

Response: Thank you for the comment. We rewritten this sentence as "*The daily maximum $T_W$ can be effectively used in monitoring humid heatwaves and their effects on health*".

Line 16 - insert "stations" after "These"

Response: Thank you for the suggestion. We have inserted the "stations" after "These".

Line 18 - its not clear at this point what the offsetting mentioned does. Is it possible to clarify a little (I appreciate it is the abstract, so space is limited).

Response: Thank you for the comment. "Offset" here means that using the GSDM-WBT could avoid the underestimation of $T_W$ calculated from reanalysis dataset. We changed the sentence as "*GSDM-WBT handles stations with many missing values and possible inhomogeneities, and also avoid the underestimation of the $T_W$ calculated from reanalysis data*".

Line 24 - "..resulting in the increase of the frequency...

Response: Thank you for the suggestion. We have changed the phrase.

line 29 - "response to"

Response: Thank you for the suggestion. We have corrected the word.

line 31 - I think it would be important to indicate here that the "relatively low air temperature" refers to heat wave conditions, I initially read this as relatively low absolute air temperatures (which for me is 0-5C).

Response: Thank you for the suggestion. We have changed this sentence as "*For example, extreme humid-heat combining with low air temperature but a high humidity might still cause*

*lethal and even deadly events*".

line 35 - Im not sure this is the representation of the WBT on its own. However my understanding is that, combined with biophysical metrics, it can be used to determine whether sweating would still be effective.

Response: Thank you for the comment. Based on the thermodynamic definition, wet-bulb temperature is the temperature of an air parcel cooled to saturation by the evaporation of water into it. In the physiological research, sweating is considered as the main cooling mechanism under extreme heat conditions. So, the $T_W$ could also be used as the lowest temperature reached by evaporative cooling of sweating. For better understanding, we changed this sentence according to your suggestion as "...*the higher $T_W$ could dampen the evaporative cooling of sweating*".

Figure 1 - include in the caption a statement saying that the numbers in parentheses are the station counts left at each stage.   Im not sure that "Correlationship" is a word (though I get what youre trying to say).   The arrow from the Complemented se ries back to Climatol is missing a head.

Response: Thank you for the suggestions. We have added the statement of the numbers in the parentheses. We changed the "Correlationship" to "Correlations". We added the arrow from the "Complemented series" to "Climatol 3.1.2".

[Figure]

Figure 1. Procedures of producing global daily maximum wet-bulb temperature (GSDM-WBT) dataset. The numbers in the parentheses indicate the counts of stations remained after each procedure.

Line 82 - add suitable reference for the ISD (e.g. Smith et al, 2011)

Response: Thank you for the suggestion. We added the citation of ISD here, as Smith et al., 2011.

Line 88 - how did you account for the shift to/from summer time when using the local time for the stations?

Response: Thank you for the comment. In this study, the local time was adjusted based on the time zone calculated from the longitude of each station. We did not consider the shift of the summer time and winter time. One main reason is that this shift is not applied all over the world, for example, there is no daylight saving time in China. Furthermore, the impact of one-hour transition between summer time and winter time could be generally low because we controlled our data as at least one $T_W$ every six hours per day. But the time zones ($\pm 12h$) are quite important to change the distributions of sub-daily $T_W$ in UTC time to the real diurnal variations.

Line 121 - there is no section 3.1.1

Response: Thank you for the comment. We changed "section 3.1.1" to "section 3.1".

Line 127 - restate that the 6-hourly intervals are in local time.

Response: Thank you for the suggestion. We added the "in local time" in this sentence.

line 135 - "(of a total of..."

Response: Thank you for the suggestion. We corrected the phrase to "of a total of…".

line 137 - "contain" (delete "s")

Response: Thank you for the suggestion. We corrected the "contains" to "contain".

line 139 - insert a line break before "According"

Response: Thank you for the suggestion. We inserted a line break here.

line 156 - replace "for" with "so that there were"

Response: Thank you for the suggestion. We changed "for" to "so that there were".

line 160 - perhaps rephrase as "...depends on how many of the surrounding stations had missing data at this step"

Response: Thank you for the suggestion. We have rewritten the phrase as "*...depends on how many missing data of surrounding stations at this step*".

line 165 - "needed"

Response: Thank you for the suggestion. We corrected "need" to "needed".

Figure 2 - can you adjust the colourmap so that it diverges when centred around 0. I appreciate there are few stations which have a bias of <0 but using a symmetrical scale will make that clearer than currently where blue-colours cover negative values and those 0-0.15C.

Response: Thank you for the suggestion. We redrew the figure by adjusting the colourmap.

[Figure]

Figure 2. Sensitivity of air pressure on $T_W$. Sensitivity, or average bias, was calculated by subtracting the daily maximum $T_W$ based on long-term average pressure by daily maximum $T_W$ calculated from sub-daily pressure. Sub-plot showed the histogram of average bias when average daily maximum $T_W$ was more than 20°C, where the red dashed line indicated the mean (0.04°C).

line 241 - move "in theory" to the front of that clause, to before "the corrected series"

Response: Thank you for the suggestion. We moved the "in theory" to the front of "the corrected series".

line 262 - please check this sentence as there cannot be more than 30/31 missing days in a month.

Response: Thank you for the comment. Here the missing days were counted in each month during all forty years, so the total of days in each month for each station is about 1200 days. For better understanding, we have revised this sentence to "*The median number of missing days in each month over past forty years in the Northern Hemisphere is less than 100 days…*".

line 272 - please add a suitable reference for the PHA algorithm

Response: Thank you for the suggestion. We added the related citation as "Menne and Williams,

2009".

Figure 3 - similar to Figure 2, please can you adjust the colourmap so that it diverges when centred around zero. Currently blue and pink colours are both negative, which could be confusing.

Response: Thank you for the suggestion. We thought you pointed out the problem of Figure 6, and we redrew this figure by adjusting the colourmap.

[Figure]

Figure 6. Average bias between daily maximum $T_W$ of GSDM-WBT and HadISD-Humidity.

line 282 - please add a suitable reference for ERA5

Response: Thank you for the suggestion. We added the citation as "Hersbach et al., 2020".

line 283 - add parentheses around "2021" in the Yan et al reference.

Response: Thank you for the suggestion. We have added the parentheses.

Figure 7 - similar to Figure 2, please adjust the colourmap to center the divergence around 0. Further guidance on this can be found at https://colorbrewer2.org. Currently it I found it difficult to identify the high and low bias regions.

Response: Thank you for the suggestion. We redrew the figure by adjusting the colourmap.

[Figure]

Figure 7. Average bias between station-based daily maximum $T_W$ of GSDM-WBT and that of the nearest grid points in HiTiSEA.

line 314 - replace "index" with "indices"

Response: Thank you for the suggestion. We have replaced "index" with "indices".

line 351 - perhaps replace "cognizing" with "characterizing"

Response: Thank you for the suggestion. We have replaced "cognizing" with "characterizing".

Figure S1 - add the number of valid and invalid stations to the caption. The grey crosses are hard to see at first glance.

Response: Thank you for the suggestion. We added the number of valid and invalid stations to the caption and also redrew the figure.

[Figure]

Figure S1. Spatial patterns of valid stations selected by data quality control.

**References:**

Hersbach, H., Bell, B., Berrisford, P., Hirahara, S., Horányi, A., Muñoz-Sabater, J., Nicolas, J., Peubey, C., Radu, R., Schepers, D., Simmons, A., Soci, C., Abdalla, S., Abellan, X., Balsamo, G., Bechtold, P., Biavati, G., Bidlot, J., Bonavita, M., De Chiara, G., Dahlgren, P., Dee, D., Diamantakis, M., Dragani, R., Flemming, J., Forbes, R., Fuentes, M., Geer, A., Haimberger, L., Healy, S., Hogan, R. J., Hólm, E., Janisková, M., Keeley, S., Laloyaux, P., Lopez, P., Lupu, C., Radnoti, G., de Rosnay, P., Rozum, I., Vamborg, F., Villaume, S., and Thépaut, J.-N.: The ERA5 global reanalysis, Q J R Meteorolog Soc, 146, 1999–2049, https://doi.org/10.1002/qj.3803, 2020.

Menne, M. J. and Williams, C. N.: Homogenization of temperature series via pairwise comparisons, J Climate, 22, 1700–1717, https://doi.org/10.1175/2008JCLI2263.1, 2009.

Mora, C., Dousset, B., Caldwell, I. R., Powell, F. E., Geronimo, R. C., Bielecki, C. R., Counsell, C. W. W., Dietrich, B. S., Johnston, E. T., Louis, L. V., Lucas, M. P., McKenzie, M. M., Shea, A. G., Tseng, H., Giambelluca, T. W., Leon, L. R., Hawkins, E., and Trauernicht, C.: Global risk of deadly heat, Nature Clim Change, 7, 501–506, https://doi.org/10.1038/nclimate3322, 2017.

Smith, A., Lott, N., and Vose, R.: The Integrated Surface Database: Recent Developments and Partnerships, Bull Am Meteorol Soc, 92, 704–708, https://doi.org/10.1175/2011BAMS3015.1, 2011.

Wang, P., Yang, Y., Tang, J., Leung, L. R., and Liao, H.: Intensified humid heat events under global warming, Geophys Res Lett, 48, e2020GL091462, https://doi.org/10.1029/2020GL091462, 2021.

**Reviewer 3**

**Comments:** Wet bulb temperature is of great significance for the study of global or regional extreme heat events and humid heat, and this study realizes the calculation and homogenization of wet bulb temperatures at 1834 sites in a global long-term sequence, which is very meaningful work. Overall, this manuscript is clear and well-written and presents interesting research, but some concerns need to address before acceptance and publication. (Minor revision)

Response: Thank you for recognizing our work and providing the comments. All responses to your comments are as follows.

1. Figure S2 is difficult for readers to distinguish different zones. I suggest using different colors and symbols to distinguish different zones

Response: Thank you for the suggestion. We redrew the figure using different colors and symbols.

[Figure]

Figure S2. Spatial patterns of 41 station zones (total 1834 stations) based on Koppen-Geiger climate classifications. Each station zone contains at least 5 stations for better homogenization.

2. As mentioned by the author, the air temperature, specific humidity, and surface pressure data of each site are reanalysis meteorological data of the nearest grid point directly extracted, but the spatial resolution of NCEP-DOE data is relatively low. When using the nearest neighbor algorithm to extract the temperature, humidity, and surface pressure data of multiple adjacent sites may be the same. Will this affect WBT calculation? If the bilinear interpolation algorithm is used for extraction, does the result of WBT change greatly?

Response: Thank you for the comments. We are sorry for the unclear descriptions on extracting reanalysis series based on NCEP-DOE dataset. Due to the relatively coarse resolution of

reanalysis data, one grid might involve two or more stations spatially. Therefore, to remove the effect caused by the same reanalysis series, we deleted the duplicate series and paired it with the station-based series with highest correlation coefficients for further bias correction. We have added related descriptions in the Methods in the revised manuscript.

We did not apply the interpolation algorithm to improve the spatial resolutions because it is hard to determine the best scales of grids due to the unevenly distributed stations. The bias correction was also introduced to get the eventual complementary series so as to reduce the uncertainties from reanalysis dataset. In addition, it is notable that the process of complementing reanalysis series was essential but with relative low impact on the whole results. As shown in Table R1, the percentages of void time steps in all stations (0.04%-2.59%) relative to 14610 total time steps were low. The analysis about the effect of complementary series will be added in the Results.

Table R1. The effect of complementary series in different station zones.

| Station zone | Number of complementary series | Number of all stations | Number of void time steps in all stations | Percentage of void time steps (%) in all stations |
|---|---|---|---|---|
| Z1 | 1 | 5 | 27 | 0.18 |
| Z2 | 1 | 8 | 8 | 0.05 |
| Z3 | 1 | 9 | 14 | 0.10 |
| Z4 | 1 | 5 | 36 | 0.25 |
| Z5 | 5 | 54 | 12 | 0.08 |
| Z6 | 1 | 9 | 10 | 0.07 |
| Z7 | 9 | 87 | 5 | 0.03 |
| Z8 | 1 | 11 | 8 | 0.05 |
| Z9 | 1 | 12 | 378 | 2.59 |
| Z10 | 45 | 451 | 6 | 0.04 |
| Z11 | 2 | 20 | 8 | 0.05 |
| Z12 | 1 | 12 | 6 | 0.04 |
| Z14 | 3 | 31 | 15 | 0.10 |
| Z15 | 4 | 35 | 12 | 0.08 |
| Z16 | 1 | 8 | 24 | 0.16 |
| Z17 | 1 | 9 | 27 | 0.18 |
| Z18 | 4 | 41 | 7 | 0.05 |
| Z20 | 1 | 5 | 169 | 1.16 |
| Z21 | 2 | 18 | 9 | 0.06 |
| Z22 | 3 | 34 | 13 | 0.09 |
| Z23 | 19 | 187 | 6 | 0.04 |
| Z24 | 4 | 41 | 7 | 0.05 |
| Z27 | 6 | 64 | 12 | 0.08 |
| Z28 | 1 | 5 | 67 | 0.46 |
| Z30 | 6 | 56 | 7 | 0.05 |
| Z31 | 4 | 38 | 35 | 0.24 |
| Z32 | 1 | 12 | 46 | 0.31 |
| Z33 | 1 | 8 | 41 | 0.28 |
| Z34 | 4 | 36 | 5 | 0.03 |
| Z35 | 1 | 7 | 145 | 0.99 |
| Z36 | 1 | 7 | 123 | 0.84 |
| Z37 | 1 | 9 | 21 | 0.14 |
| Z38 | 1 | 8 | 21 | 0.14 |
| Z39 | 2 | 24 | 41 | 0.28 |
| Z40 | 1 | 9 | 32 | 0.22 |
| Z41 | 1 | 11 | 17 | 0.12 |

3.  Line 168: How is the initial daily maximum WBT calculated? The time resolution of
    NCEP-DOE reanalysis data is 6h instead of 1h. More detailed description is required here.

Response: Thank you for the comment and suggestion. We at first calculated sub-daily (6h intervals) wet-bulb temperature by the same algorithm as demonstrated in section 2.2 and computed the daily maximum wet-bulb temperature, which is the initial wet-bulb temperature before bias correction. The six-hour interval of NCEP-DOP was also consistence with the criteria of our data quality control in section 2.3. We added more details here for better understanding, and also emphasized the time resolution again in the manuscript. The revised sentences are as follows "*First, the air temperature, specific humidity and surface pressure of the grid point nearest to each station were extracted, and the sub-daily (six-hour interval) $T_W$ was calculated (see section 2.2). Then the initial series of daily maximum $T_W$ and monthly mean were computed before bias correction.*".

4. More detailed description of Figure 4 (a) is helpful for readers to compare the results before and after homogenization. In addition, "Before homogeization" should be "Before homogenization" in Figure 4 (a).

Response: Thank you for the suggestions. Figure 4 (a) showed the correlation coefficients of series between paired stations before and after homogenization, and the sub-plot showed for the paired stations of which distances lower than the first quarter. When the coefficients were more than 0, the dots in the upper areas of black diagonal indicated the higher coefficients after homogenization. The maximum increment of coefficients was 0.28. There was also an obvious increase in the coefficients between closer stations as shown in the blue dots. In the sub-plot of Figure 4 (a), about 80.23% of paired stations had larger coefficients after homogenizations. We have added more descriptions in the Results as "…*Overall, the correlation coefficients after correction were higher and the maximum increment of coefficients was 0.28. It is also notable that there was a significant increase in correlation between stations that were closer together as shown in the blue dots. In the sub-plot of Figure 4 (a), about 80.23% of paired stations had larger coefficients after homogenizations…*", and in the caption of figure as "*When the coefficients were more than 0, the dots in the upper areas of black diagonal indicated the higher coefficients after homogenization*". We also revised "Before homogeization" to "Before homogenization" in Figure 4(a).

[Figure]

Figure 4. Correlation coefficients (p<0.05) between paired series before and after homogenization (a), annual average daily maximum $T_W$ (℃) and the number of infilled or corrected data for one typical station in each station zone (Z1-Z41). Note that sub-plot of (a) showed the correlation coefficients between paired stations of which distances lower than the first quarter. When the coefficients were more than 0, the dots in the upper areas of black diagonal indicated the higher coefficients after homogenization. Detailed information of all typical stations was shown in Table S2.

5.  Line 243: What does higher SNHT value represent?

Response: Higher SNHT values mean higher probability of such stations to be detected the break points (also the inhomogeneous series). In the Climatol, SNHT test is applied to the series of anomalies between the actual values and the reference values and the SNHT values are used

to identify the break points. We have further added the related explanation in the Methods as "*Higher standard deviations and SNHT values mean higher probability of such stations to be detected as the outliers and break points.*".

6. In Lines 264-265: The WBT is calculated site by site and day by day. The statistical results do show that there are many missing data of WBT in the HadISD-Humidity data, but the author believes that HadISD-Humidity has relatively low accuracy and higher uncertainties. From my understanding, the existing description is not enough to prove that HadISD-Humidity has relatively low accuracy and larger uncertainties.

Response: Thank you for the comments. In this section, we aimed to explain two main problems in HadISD-Humidity, and the first one is about missing values. Particularly, since the daily maximum $T_W$ is the main measurement for characterizing extreme humid-heat in the warm seasons, the number of missing days in different months was shown in the manuscript. It was found that there were more missing values during the warm season, especially in the Northern Hemisphere. Because the extremely humid heat events are generally identified based on daily $T_W$ and the daily thresholds in the historical baselines, more missing values could lead us to detect inaccurate thresholds or identify insufficient events. So, the probable uncertainties may exist when directly using HadISD-Humidity to characterize humid heat.

We are sorry for the misunderstanding on the description, and according to the above explanation we revised the related contents as "*In terms of seasonality, there are evidently more missing days in the warm season (May-September) in the Northern Hemisphere, especially in summer (June-August). Because the extremely humid heat events are generally identified based on daily $T_W$ and the daily thresholds in the historical baselines, more missing values could cause inaccurate thresholds or insufficient events to be detected. Therefore, it needs to be noticed the probable uncertainties when directly using HadISD-Humidity to characterize humid heat*".

7. As described by the author, the results of HadISD-Humidity and HiTiSEA are overestimated or underestimated, and the author uses HadISD-Humidity and HiTiSEA data to compare and analyze the results of GSDM-WBT, then the verification results do not represent the true accuracy of GSDM-WBT, but only the relative accuracy.

Response: Thank you for the comments. To get the true accuracy of GSDM-WBT, the best approach is to directly compare our dataset with the long-term homogenous observations of wet-bulb temperature. However, to our knowledge, there is no global observation-based dataset which could be used to validate the absolute accuracy of GSDM-WBT until now. According to

your suggestion, we would emphasize the "relative accuracy" of the evaluations in the revised manuscript.

As for the dataset HiTiSEA, its underestimation of average 0.4°C was also found by their producers. When comparing the HiTiSEA to GSDM-WBT, the average bias was about 0.34°C which proved its underestimation. In previous regional studies on $T_W$, the underestimation caused by using reanalysis dataset has also been demonstrated (Freychet et al., 2020; Raymond et al., 2020). When comparing with the HadISD-Humidity, the daily maximum $T_W$ of GSDM-WBT is overall lower. Because of the same data source used in the GSDM-WBT and HadISD-Humidity, the existing missing values might increase the potential inhomogeneity of HadISD-Humidity. So, in the section of comparing with station-based dataset, we would like to illustrate the possible reasons for the bias between GSDM-WBT and HadISD-Humidity, but not determine whether the HadISD-Humidity was overestimated. We have checked all inappropriate descriptions and revised the contents.

**References:**

Freychet, N., Tett, S. F. B., Yan, Z., and Li, Z.: Underestimated change of wet-bulb temperatures over east and south China, Geophys. Res. Lett., 47, e2019GL086140, https://doi.org/10.1029/2019GL086140, 2020.

Raymond, C., Matthews, T., and Horton, R. M.: The emergence of heat and humidity too severe for human tolerance, Sci. Adv., 6, eaaw1838, https://doi.org/10.1126/sciadv.aaw1838, 2020.

**Community comment**

**Comments:** Thank you for considering many factors to produce a quality controled wet bulb temperatue (Tw) dataset. In framing the discussion in reference to heat stress: when Tw is higher than >20°C, Tw becomes relevant for human and animal health. Something to think about might be checking the effect of long-term pressure on high Tw, because small changes on high Tw can produce large changes in impacts. Are the biases stable at high Tw?

Response: Thank you for the comment. The effect of long-term average air pressure was analyzed in the manuscript section of Sensitivity Analysis. As for high wet-bulb temperature, the average bias for such stations where average daily maximum $T_W$ was above 20°C was around 0-0.11°C, which is similar with previous studies such as Raymond et al., 2020. We added the related description in the revised manuscript as "*When targeting on the stations with average daily maximum $T_W$ more than 20°C, where humid heat conditions are highly relevant to human health, the average bias was also maintained at 0-0.11°C*", and the histogram of such average bias was also shown in the Figure 2.

[Figure]

Figure 2. Sensitivity of air pressure on $T_W$. Sensitivity, or average bias, was calculated by subtracting the daily maximum $T_W$ calculated by sub-daily pressure from the daily maximum $T_W$ based on long-term average pressure. Sub-plot showed the histogram of average bias when average daily maximum $T_W$ was more than 20°C, where the red dashed line indicated the mean (0.04°C).

I also appreciate referencing the HumanIndexMod. There is an updated version which is python compatible correcting a minor error in the code. The error is <±0.001 in Tw. (https://github.com/jrbuzan/HumanIndexMod_2020). This uses f2py as part of numpy to compile the fortran object. On top of that, unlike Dr. Kopps matlab version, this python enabled

version will give you access to all of the heat stress algorithms in the code block, not only Tw. Convient for compairing Tw with commonly used heat indices.

Response: Thank you for the comment. We noticed the minor error of old versions code and also referred the updated version to calculate $T_W$ in this study. It is quite useful to use HumanIndexMod to calculate different heat indices. However, as described in the manuscript, the new dataset not only focused on the calculation of $T_W$. It is also expected to obtain the better quality-controlled and homogenous data which needs a series of complex procedures. Therefore, we decided to choose one important and representative indicator to produce the dataset for supporting the research of humid heat events. The wet-bulb temperature, a thermodynamic variable which has been globally used and recognized to analyze the humid heat stress and its effect, was eventually selected. It is also hoped that more datasets including other heat indices could be produced in the future.

A pedantic, but necessary comment on acronymes, abbreviations, and subscripting of heat stress indexes vs thermodynamic state variables. Wet bulb temperature is a true thermodynamic property of the atmosphere. It reflects the bouyancy of the air, and is directly related to equivalent potential temperature, and as thus, is an atmospheric state variable. Since wet bulb temperature is intrinsically a form of temperature, it is denoted with a subscript, for example using the LaTeX code: $T_{w}$. This differentiates wet bulb temperature from heat stress indices, like the Universal Thermal Climate Index, which is abbreviated as UTCI, HUMIDEX, or Apparent Temperature (AT). The heat stress indices are not true thermodynamic state variables, which is why they are traditionally abbreviated as capital letters. This is an easy fix, and would be in line with atmospheric, medical, and epidemiological literatures. The acronym WBT was recently introduced into the literature, and i believe it is due to using the raw output labled as WBT from CLM5 netcdf files, and authors not converting it to the state variable, Tw.

Response: Thank you for the suggestion. We have changed all "WBT" in the texts and figures to "$T_W$" to represent wet-bulb temperature. But we did not change the name of our dataset, considering that "GSDM-WBT" could still better involve the initials of "Global Station-based Daily Maximum Wet-bulb Temperature".

Lastly, I encourage the authors to check out Buzan and Huber, 2020. I describe in the article why sub-daily calculations of heat stress related variables is important, which is relevant to the motivation for in your work here using sub-daily values. Also, there are many insights in that manuscript that one might find fascinating on the topic of moist heat stress that may be useful

in your discussion section. Additionally, Poppick and McKinnon 2020 and McKinnon and Poppick 2020 are likewise important manuscripts describing the statistical robustness behind temperature-humidity covariances, and would be good additions to your discussion.

Response: Thank you for the suggestion. Buzan and Huber (2020) found that the differences of extreme humid heat between using monthly and sub-daily temperature and humidity could be up to more than 4°C at reginal scales, which provided an important proof of why we produced the daily maximum $T_w$ based on the sub-daily data sources. Related description as "*Previous research showed the differences of extreme humid heat between using monthly and sub-daily temperature and humidity could be up to more than 4°C at reginal scales, and lead to substantial effect on future predictions (Buzan and Huber, 2020)*" was added in the revised manuscript.

Poppick and McKinnon (2020) and McKinnon and Poppick (2020) investigated the changing relationship between temperature and humidity using a flexible statistical method, and provided the new sights for simulating the future temperature and humidity, thus expanding our dataset in the future. We added more contents in the Discussion as "*Meanwhile, the complex changes in the relationship but not the simply fixed joint between temperature and humidity, was investigated around different regions based on the multivariate analysis (Mckinnon and Poppick, 2020). Then the historical dataset of $T_W$ could be expanded to future longer periods based on observation-based relationship between temperature and humidity (Poppick and Mckinnon, 2020)*".

**References:**

Buzan, J. R. and Huber, M.: Moist heat stress on a hotter earth, Annu Rev Earth Planet Sci, 48, 623–655, https://doi.org/10.1146/annurev-earth-053018-060100, 2020.

McKinnon, K. A. and Poppick, A.: Estimating changes in the observed relationship between humidity and temperature using noncrossing quantile smoothing splines, JABES, 25, 292–314, https://doi.org/10.1007/s13253-020-00393-4, 2020.

Poppick, A. and Mckinnon, K. A.: Observation-based simulations of humidity and temperature using quantile regression, J Climate, 33, 16, https://doi.org/10.1175/JCLI-D-20-0403.1, 2020.

Raymond, C., Matthews, T., and Horton, R. M.: The emergence of heat and humidity too severe for human tolerance, Sci. Adv., 6, eaaw1838, https://doi.org/10.1126/sciadv.aaw1838, 2020.